# Dielectric Characterization of Ex-Vivo Breast Tissues: Differentiation of Tumor Types through Permittivity Measurements

**DOI:** 10.3390/cancers16040793

**Published:** 2024-02-15

**Authors:** Elizabeth G. Fernández-Aranzamendi, Patricia R. Castillo-Araníbar, Ebert G. San Román Castillo, Belén S. Oller, Luz Ventura-Zaa, Gelber Eguiluz-Rodriguez, Vicente González-Posadas, Daniel Segovia-Vargas

**Affiliations:** 1Department of Signal Theory and Communications, University Carlos III of Madrid, 28911 Madrid, Spain; esanroman@ucsp.edu.pe (E.G.S.R.C.); belsegoller@gmail.com (B.S.O.); vicente.gonzalez@upm.es (V.G.-P.); 2Department de Ingeniería Eléctrica y Electrónica, Universidad Católica San Pablo, Arequipa 04001, Peru; pcastillo@ucsp.edu.pe; 3Department of Oncology Medicine, Regional Institute of Neoplastic Diseases, Arequipa 04002, Peru; luzventurazaa@gmail.com (L.V.-Z.); eguiluz24@hotmail.com (G.E.-R.)

**Keywords:** breast, tumor, radiofrequency, coaxial probe, measurements, dielectric characterization

## Abstract

**Simple Summary:**

Breast cancer, a prevalent global concern, continuously motivates the research and development of new technologies for its quick detection and diagnosis. Dielectric spectroscopy is a technique previously used for material characterization but with limitations in accuracy for breast cancer diagnosis due to a lack of control over the material to be characterized. For the first time, and thanks to the assistance of 70 volunteer patients, accurate tissue characterization of breast tumors, benign or malignant (ductal carcinoma, lobular carcinoma, mucinous carcinoma and fibroadenoma), was achieved with dielectric spectroscopy. The results obtained in both relative permittivity and loss tangent show, for the first time, notable differences between the four types of tumors, which can help in making an instantaneous diagnosis of the excised tumor and accelerating subsequent treatment.

**Abstract:**

Early analysis and diagnosis of breast tumors is essential for either quickly launching a treatment or for seeing the evolution of patients who, for instance, have already undergone chemotherapy treatment. Once tissues are excised, histological analysis is the most frequent tool used to characterize benign or malignant tumors. Dielectric microwave spectroscopy makes use of an open-ended coaxial probe in the 1–8 GHz frequency range to quickly identify the type of tumor (ductal carcinoma, lobular carcinoma, mucinous carcinoma and fibroadenoma). The experiment was undertaken with data from 70 patients who had already undergone chemotherapy treatment, which helped to electrically map the histological tissues with their electric permittivity. Thus, the variations in the permittivity of different types of tumors reveal distinctive patterns: benign tumors have permittivity values lower than 35, while malignant ones range between 40 and 60. For example, at a frequency of 2 GHz, the measured permittivity was 45.6 for ductal carcinoma, 33.1 for lobular carcinoma, 59.5 for mucinous carcinoma, and 27.6 for benign tumors. This differentiation remains consistent in a frequency range of 1 to 4.5 GHz. These results highlight the effectiveness of these measurements in the classification of breast tumors, providing a valuable tool for quick and accurate diagnosis and effective treatment.

## 1. Introduction

Statistics from the Global Cancer Observatory for the year 2020 revealed that breast cancer became the most diagnosed type of cancer worldwide, with more than 2.26 million new cases and nearly 685,000 deaths related to breast cancer globally [1]. Reference source not found. Specifically in Peru, where we have taken this set of measurements, according to the report from the League Against Cancer for the same year, breast cancer topped the mortality statistics. A total of 6960 cases were diagnosed, representing a worrying 9.8% of all cases detected in the country [2].

These data underscore the urgency of exploring alternative approaches for the detection, diagnosis, treatment, and monitoring of breast cancer. We start by understanding the procedure when breast cancer is suspected, which begins with the use of technologies such as mammography, ultrasound, and magnetic resonance imaging for monitoring and detecting potential abnormalities in breast tissue [3]. At this moment, it is important to introduce the BIRADS protocol into the radiology process. Thus, the BIRADS system provides the radiologist with four breast categories based on breast density. Types A and B correspond to non-dense breasts, primarily composed of fatty tissue and, in some cases, scattered glandular tissue; these types are mostly associated with non-young women. The fatty tissues allow for easier detection of the potential tumor. Secondly, Types C and D refer to breasts with dense fibro-glandular tissue that is distributed heterogeneously. These types are mostly associated with young women and more muscular tissues. These types of tissues make the correct detection of the tumor more difficult; thus, subsequent diagnosis techniques must be applied.

If irregularities are detected, a biopsy is performed, and the tissues are stained with hematoxylin and eosin for microscopic analysis by pathologists, which is crucial for diagnosis. Depending on the stage of cancer, treatment may include chemotherapy or radiotherapy before determining surgery (mastectomy or lumpectomy). It is relevant to mention that most surgeries are usually preceded by chemotherapy treatments, followed by a pathological analysis of the extracted tissue to determine the tumor’s typology [4,5]. However, this process often involves long waiting times and depends on the availability of specialized technology and qualified professionals [6,7].

Histological classification helps to clearly identify the final benignant or malignant character of the tumor. This classification is based on the appearance and structure of cancer cells observed under a microscope. Among the most common histological tumor types are infiltrating ductal carcinoma (IDC), infiltrating lobular carcinoma (ILC), ductal carcinoma in situ (DCIS), lobular carcinoma in situ (LCIS), inflammatory breast carcinoma, papillary carcinoma, medullary carcinoma, mucinous carcinoma, and tubular carcinoma [4,5,6,7]. As the infiltrating character is achieved by a further expansion of the corresponding in situ tumor, from the electrical characterization point of view, we will focus on ductal carcinoma, lobular carcinoma, mucinous carcinoma, and fibroadenoma or benign tumors. Finally, the papillary, medullary and tubular carcinoma are more unusual tumors and were not found in the 70 volunteer patients. For this reason, Table 1 shows the following four types of tumors: ductal, lobular, mucinous, and fibroadenoma.

However, emerging technologies such as the application of electromagnetic fields can provide an additional diagnostic and monitoring avenue for cancer-related diseases. This technology is not intended to replace existing ones but to complement them by offering a prediagnosis. The interaction of electromagnetic (EM) fields with the human body is directly related to the inherent dielectric properties of each tissue. These properties determine how biological tissues transmit, absorb, and reflect electromagnetic waves in varying proportions. In general terms, tissues with high proportions of polar molecules, primarily water, tend to have high permittivity. Conversely, nonpolar compounds, like adipose tissues, exhibit low permittivity [8,9,10,11,12,13,14]. These characteristics provide valuable information for identifying unwanted pathologies such as breast cancer, among other applications [15,16,17,18,19,20,21,22]. Various research groups have focused on leveraging tissue contrast between benign and malignant breast cancer for many years, and recently several systems have been implemented in clinical practice. However, no studies have shown histological variations in mammary tumors. At a fundamental level, this contrast is largely due to water, as the predominant adipose tissue has much lower dielectric properties than tumors with higher water content [23,24,25,26]. Nevertheless, more complex analyses have suggested that mechanisms such as bound water effects could also play a role in specific frequency ranges [24,25,26,27,28]. Although characterization of breast tumor tissue has been carried out, histological classification of these tissues has not been performed. This is due to the inaccuracy of the method, influenced by factors such as the sensitivity of the method, external environmental conditions, and the interface between the probe and the sample. These elements are crucial when applying methods such as the open circuit coaxial probe. Among the environmental factors of importance to consider are temperature, pressure, and the sample’s exposure time from excision; in addition to this [29,30,31], it is necessary to consider the mammary tumor’s structure [32,33].

This analysis of properties is fundamental for various applications, such as microwave imaging techniques for the early detection of breast cancer, as well as for treatments and diagnostics [33,34,35,36,37,38,39]. Therefore, extensive studies have been carried out for many decades to explore the multiple potential uses. However, the accuracy of this technology is influenced by an adequate dielectric characterization of breast and tumor tissues.

In response to the identified needs, the goal of this article is to perform a histological dielectric characterization of breast tumor tissues. For this, a protocol with a controlled measurement system is employed. These measurements are carried out when the patient is in the last phase of the medical procedure, which is determined by the doctor’s diagnosis after having completed chemotherapy treatment and being scheduled for surgery. The purpose is to provide essential knowledge for electromagnetic detection and characterization techniques. In designing the experiment, an open coaxial probe was used in a frequency range of 1 to 8 GHz, implementing a measurement protocol that controls variables such as pressure, time, and temperature. The data collected from 70 patients were classified into four groups based on the type of tissue. Four different models were used for dielectric characterization, the results of which were later compared with the pathological analysis of the samples.

The paper is organized into six sections. The first two (0 and 1) detail the summary and introduction, as presented. The second provides a detailed description of the materials and methods used. The third presents the results obtained from the dielectric characterization. The fourth shows the differences observed between the types of tissues analyzed based on the results. The fifth discusses the results of the article and finally, the sixth section contains the conclusions derived from the study.

## 2. Materials and Methods

This study was conducted at the Surgery Unit of the Regional Institute of Neoplastic Diseases of Southern Peru, with the approval of the Bioethics Committee. All participants provided informed consent to access their medical records and the results of the examinations of the excised tissues, which were exclusively used for research purposes. The research included a total of 70 patients who underwent lumpectomies, partial mastectomies, and radical mastectomies after having had chemotherapy treatment. From these patients, between 1 and 3 samples were obtained per individual, totaling 279 samples of different histological types, including benign tissue.

The measurement method used was non-destructive, allowing for subsequent pathological analysis of the same sample. A protocol was designed that minimized handling and the elapsed time between removal and measurement, all within a controlled environment system. This system could measure pressure and rotate the sample; additionally, ambient temperature was manually recorded.

As the goal of the paper is the electromagnetic characterization of the various breast tumor types, the decision we have made is to work with the women who could have more various and aggressive tumors. The medical protocol establishes three kinds of processes: lumpectomy for small and non-aggressive tumors, mastectomy for extensive but non-aggressive tumors, and chemotherapy and mastectomy for extensive and aggressive tumors. Thus, the third situation was the most likely to have access to all types of tumors. For this reason, the chosen patients underwent chemotherapy before tumor excision.

It is important to emphasize that chemotherapy can significantly alter breast tissues. If a sample is extracted during or immediately after chemotherapy treatment, the measurements might not reflect the actual dielectric behavior of the tissue as it is under the effects of the treatment. Therefore, to ensure the accuracy of our observations, all measurements were conducted after the patients had completed their chemotherapy cycle. This ensures that the results obtained more accurately reflect the dielectric behavior of the breast tissue under equal conditions.

In the following sections, we begin by providing an overview of breast cancer and the conventional methods of tissue classification used in clinical practice. Then, we detail the protocol used to measure the complex permittivity of freshly excised breast samples using an open coaxial probe, followed by an analysis and classification of the results obtained.

### 2.1. Sample Classification

Abnormal growth in breast tissues is known as a tumor. These tumors can be benign (non-cancerous) or malignant (cancerous). For the diagnosis and treatment of breast cancer in daily clinical practice, various methods are used to classify and characterize tissues. Diagnostic procedures include ultrasound, mammography, and magnetic resonance imaging, which provide images with contrast differences, allowing visualization of the tumor.

However, the only method that can definitively determine whether a tumor is cancerous is pathological examination. Through this, the tissue is analyzed and characterized, identifying the type of tumor present. After this diagnosis, an appropriate treatment is defined, including surgery and/or chemotherapy. If removal is chosen, it is essential to perform a new pathological examination of the extracted sample to accurately determine the nature of the anomaly and thus have a complete medical record.

As previously stated, the electrical characterization will be based, as seen in Table 1, on ductal carcinoma, lobular carcinoma, mucinous carcinoma, and fibroadenoma. The importance of histological classification lies in its direct influence on therapeutic decisions and the prognosis of the disease. Each variant of breast cancer has different management and treatment, adapted to its histological type and the individual characteristics of the patient. In this article, we will focus on three histological types of malignant tumors, as well as one benign type, shown in Figure 1.

### 2.2. Measurement System

Figure 2 shows the setup for taking measurements of the breast tissues and drawing out the dielectric properties. The setup consists of the following elements:A rotating circular tableThe DAK 3.5(TL2) probeA vector network analyzer (VNA) (i.e., Copper Mountain S5085)Computer software for the DAK 3.5

All the items can be described as follows. The first is an automated control system that allows for contact and contactless measurements (Element 1). It has been updated to be able to apply a preset pressure while offering the capability to rotate based on the type of required measurement. Once the automated platform has been briefly presented, the second element is the coaxial probe (a DAK 3.5 device from the company Speag), called Element 2 in Figure 2. It is based on an open-circuit coaxial probe [24,25]. As shown in Figure 3, the open end of the coaxial probe consists of a *r_dak_* radius flat platform surrounding the open-ended coaxial probe with an inner radius (*r*_1_ = 2 mm) and an outer radius (*r*_2_ = 3 mm). The sensing fundamental is based on the variation of the ending capacity of the probe. The capacity per unit length is given by
(1)$$C_{t}=\frac{2\pi \varepsilon }{ln(\frac{r_{2}}{r_{1}})}$$

According to (1), the variation in the relative permittivity makes the measured capacitance change. Once the probe is defined, the next step is to put it in contact with the tissue sample. The overall measurement structure results in a multilayer model, as seen in the right part of Figure 3. Then, the new ending capacitance is modified according to the tissues in contact. As can be seen in Figure 3, the new resulting capacitance *C_t_* will be the result of the shunted connection of the forward capacitance of the probe, *c_f_*, and the subsequent shunted capacitances, *c*_1_ and *c*_2_, formed between the probe and the corresponding tissue layers. This is represented as follows:(2)$$C_{t}=c_{f}+c_{1}+c_{2}$$

The following step is the probe calibration that uses the OPEN, SHORT, and WATER calibration system. Then, the permittivity calculation is based on the comparison between the calibrated data and the ending capacitance and impedance variation associated with the tissue variation.

The third element (3 in Figure 2) is the Copper Mountain S5085 vector network analyzer (VNA). For this system, the S parameters are used, making it essential to connect with the Dak 3.5. Lastly, Element 4 represents the computer, which is utilized to run the Speag program and obtain the permittivity from the measured S-parameter. This software operates on an algorithm grounded in the Debye equations and refined through numerical methods. As a result, dielectric parameters are determined. The research primarily focuses on analyzing data such as real permittivity, conductivity, and loss tangent.

### 2.3. Protocol

The measurement protocol depicted in Figure 4 consists of five crucial steps: preliminary preview, instrument preparation, ex vivo procedure, data analysis, and pathological diagnosis.
Preliminary review: This stage involves assessing various prior monitoring results, including ultrasound, mammography, and medical diagnoses. The exact location of the tumor is identified and documented thanks to the surgeon’s help. The tumor size is determined and noted.Instrument preparation: The vector network analyzer (VNA) is turned on 30 min before the measurement to stabilize thermal effects. This ensures the equipment is stabilized and ready for use. Calibration of the Speag DAK equipment is carried out using a standard open-short and water method. This step is vital to guarantee accurate measurements.Ex vivo procedure: The pressure is verified, ensuring a measure of 22 kPa calculated based on the measuring instrument that will be in contact with the tissue. This ensures the tissue retains its shape characteristics while maintaining sufficient contact for measurement. Ambient temperature is recorded, noting specific values between 22° and 27 °C for correct estimation of the electrical characteristics of the tumor. The exact time from tissue removal to measurement is determined to be a maximum of 5 min. Data are collected from three different points on the same sample to ensure accuracy and provide a comprehensive view.Data analysis: With the collected data, specific values for permittivity and conductivity are computed. These metrics are vital for understanding the electrical properties of tumor tissue.Pathological diagnosis: Finally, with the measurement data in hand, they are cross-referenced with the pathology results.

## 3. Experiment Design

The experiment was structured into two fundamental phases: experimental and analytical. The experimental phase aimed to perform dielectric measurements on real tissues, using the equipment specified in Figure 2. It is important to note that all excised patients had previously received and completed their chemotherapy treatment. This choice was made for patients who had very aggressive tumors and where the chemical-directed tracer could not be properly established. Under these conditions, the presence of notable variations attributable to chemotherapy could have a significant impact on the experiment’s results, if administered during the ongoing treatment. In the case of benign tumors, the volunteers additionally had malignant aggressive tumors, hence they had undergone chemotherapy. Therefore, it was deemed crucial to adjust the group selection to control these variables and ensure the accuracy of the results.

The methodology of this stage is explained in Figure 5. It included “ex vivo” measurements, that is, on excised tissues, applied in two types of surgical procedures: mastectomies (partial or total removal of the breast) and lumpectomy (removal of the breast tumor), illustrated in Figure 6. Once the excised tumors are available, they are properly placed on the rotating table under the coaxial probe to be measured with the VNA. The measurements are carried out following the protocol described in Figure 4, focusing directly on the breast tumors in both procedures. In mastectomies, the surgeon identifies and marks the tumor area, making an incision for direct measurement. In lumpectomy, the measurement is simplified due to the type of surgery involving the direct extraction of the tumor. Once the electrical characterization has been undertaken, subsequent pathology is carried out for the classification of tumors into four types: ductal carcinoma, lobular carcinoma, mucinous carcinoma (all malignant), and fibroadenoma (benign).

The second stage of the experiment is the analytical analysis. Using Matlab 2023b, the data are analyzed to evaluate both the mean and the variance of the real and imaginary permittivity, conductivity, and loss tangent. The four most common dielectric models are applied and, through the calculation of minimum error, the most suitable model is selected. This meticulous approach ensures precise analysis of the data obtained.

Table 1 provides a summary of 70 patients, classifying breast tumors based on histological reports. For cases with multiple tumors, individual measurements were taken for each tumor. It is noted that these patients had completed chemotherapy and were scheduled for surgery. The data are divided into malignant and benign tumors. For malignant tumors, 120 measurements were taken from 40 ductal carcinoma samples from 30 patients, and 39 measurements from 13 samples of lobular and mucinous carcinoma from 10 patients each. For benign tumors, 27 fibroadenoma samples from 20 patients were analyzed, totaling 81 measurements. This group included patients with malignant tumors who had completed chemotherapy treatment.

## 4. Result and Discussion

According to Figure 5 and Table 1, the described procedure provides us with 279 measurements of the breast tissues: 198 of malignant tumors (120 of ductal carcinoma, 39 of lobular carcinoma and 39 of mucinous carcinoma) and 81 of benign tumors. These measurements provide us with 279 values of the real and of the imaginary part of the permittivity at each frequency. To conduct a detailed analysis, the process involved deriving results by calculating the mean and variance from the 279 measurements taken for each sample. Thus, the mean and the variance are calculated for any of the four types of tumor tissues within the number of available measurements. The variance is determined using Equation (3), where the term *n* denotes the total number of measurements for each type of tumor tissue (120, 39, 39 or 81, respectively). The term $\bar{x}$ refers to the mean value obtained from the available measurement while *x_i_* denotes the current measurement. Additionally, the index *i*, ranging from 1 to *n*, indicates the number of repetitions conducted at the same frequency point.
(3)$$Variance=\sum_{i=1}^{n}\frac{{\left(x_{i}-\bar{x}\right)}^{2}}{n}$$

This process is repeated 71 times for 71 frequency points in the range from 1 to 8 GHz.

Figure 7 shows how both the real and imaginary permittivity vary across the four types of analyzed tumors. A decreasing trend in the permittivity value can be seen as the frequency increases in all four types of examined tissues. Mucinous carcinoma, characterized by its more liquid or aqueous composition, exhibits the highest permittivity among them. On the other hand, fibroadenoma, being a benign tumor, shows the lowest permittivity. These findings confirm that higher permittivity (above 30 in the real part and 14 in the imaginary part) is associated with malignant tumors. The bars in the graph indicate the variance over the mean values.

From the real and imaginary part of the permittivity, it is possible to derive values for the conductivity and loss tangent, which are represented in Figure 8. These parameters are crucial for distinguishing between the four types of tumors that were examined. Figure 8a demonstrates the increase in conductivity, which describes the tissue’s ability to conduct electric current, in various tumor types as the frequency increases. Within the studied frequency range, mucinous carcinoma shows the highest conductivity, while fibroadenoma has the lowest. Moreover, Figure 8b highlights how the dielectric loss factor, represented by tan(δ), varies. This factor indicates the amount of energy converted into heat when an electric field acts on a material, in this case different types of tumors as a function of frequency in Hz. It is observed that ductal carcinoma has the highest value of tan(δ) across the entire frequency range, while fibroadenoma shows slightly lower values compared to the other tumors in the same range.

It is observed that all tumors, both malignant and benign, exhibit a decrease in permittivity as the frequency increases from 1 GHz to 8 GHz. However, some confusion is noted in the range of 5 to 8 GHz, where measurements might overlap. Therefore, it is concluded that a useful range for characterization is between 1 and 4 GHz, with significant variations noted at 2 and 4 GHz frequencies. This observation allows for the selection of frequencies within this specific range to better define the behavior of breast tumors, thereby optimizing electromagnetic systems and reducing the need for more complex systems.

In terms of specific characteristics, mucinous carcinoma demonstrates the highest permittivity, while ductal carcinoma stands out for its higher loss tangent. On the other hand, benign tumors, exemplified by fibroadenoma, show the lowest permittivity, loss tangent, and conductivity compared to malignant tumors.

Table 2 presents a summary of the data gathered for four different types of tumors, focusing on the frequency range of 2 to 4 GHz. This specific interval was chosen due to the significant and characteristic variations noted among the different tissue types, in addition to consistency in the measurements without any cross-interferences. The importance of the data and the details of this analysis provide crucial elements for a deep understanding of the unique properties of each tumor type within this frequency spectrum. The findings suggest that for tumor discrimination, it might suffice to use only the 2 to 4 GHz range, which would simplify diagnostic procedures.

After collecting and comparing experimental data, a more detailed characterization of materials was conducted using four different theoretical models: the Debye Model, the Double Debye Model, the Cole–Cole Model, and the Double Cole–Cole Model. The mathematical equations representing these models are found in Equations (4)–(8). Subsequently, MATLAB was used to graph and analyze their behavior, as illustrated in Figure 9, Figure 10, Figure 11 and Figure 12. These models incorporate variables such as real and imaginary permittivity, as well as conductivity. Specifically, for the tissues analyzed, which are characterized by their heterogeneity and various relaxation times, the Double Cole–Cole Model proved to be the most accurate, exhibiting the lowest error, as detailed in Table 3.

The complex permittivity model *ε*^∗^(*f*) as a function of frequency (*f*):(4)$$\varepsilon^{*}\left(f\right)=\frac{\varepsilon^{\prime }}{\varepsilon_{0}}-j\frac{\varepsilon^{\prime \prime }(f)}{\varepsilon_{0}}$$

Debye model:(5)$$\varepsilon \left(\omega \right)=\varepsilon_{\infty }+\frac{\varepsilon_{s}-\varepsilon_{\infty }}{1+j\omega \tau }+\frac{\sigma }{j\omega }$$

The Double Debye model:(6)$$\varepsilon_{r}^{*}\left(\omega \right)=\varepsilon_{\infty }+\sum_{m=1}^{M}\frac{∆\varepsilon_{m}}{1+{\left(j\omega \tau_{m}\right)}^{1-\alpha_{m}}}+\frac{\sigma_{s}}{j\omega \varepsilon_{0}}$$

The single Cole-Cole model:(7)$$\varepsilon_{r}^{*}\left(\omega \right)=\varepsilon_{\infty }+\frac{∆\varepsilon }{1+{\left(j\omega \tau \right)}^{1-\alpha }}+\frac{\sigma }{j\omega \varepsilon_{0}}$$

The Double Cole-Cole model:(8)$$\varepsilon^{*}\left(\omega \right)=\varepsilon_{\infty }+\frac{∆\varepsilon_{1}}{1+{\left(j\omega \tau_{1}\right)}^{1-\alpha_{1}}}+\frac{∆\varepsilon_{2}}{1+{\left(j\omega \tau_{2}\right)}^{1-\alpha_{2}}}+\frac{\sigma }{j\omega \varepsilon_{0}}$$
where$\varepsilon^{*}(f)$ is the complex permittivity as a function of frequency.*ε*′ is the real part of permittivity.*ε*″(*f*) is the imaginary part of permittivity as a function of frequency.$\varepsilon_{0}$ is the permittivity of free space.*j* is the imaginary unit.$\varepsilon_{r}^{*}\left(\omega \right)o\varepsilon \left(\omega \right)$ is the relative complex permittivity as a function of angular frequency *ω*.$\varepsilon_{\infty }$ is the permittivity at infinite frequency.$\varepsilon_{s}$ is the static permittivity in the low frequency limit, representing the maximum polarization of the material.∆ε and $∆\varepsilon_{m}$ are the increments of permittivity.τ and $\tau_{m}$ are the relaxation times.*α* and $\alpha_{m}$ are the dispersion parameters.*σ* and $\sigma_{s}$ are the conductivities.*M* is the number of terms in the summation (for the Double Debye model).$∆\varepsilon_{1}$, $∆\varepsilon_{2}$, $\tau_{1}$, $\tau_{2}$, $\alpha_{1}$, $\alpha_{2}$ are changes in permittivity, relaxation times, and dispersion parameters for two different relaxation processes.*j* is the imaginary unit *j*2 = −1.$\varepsilon_{0}$ is the permittivity of free space.

Equation (9) for the error is a sum of the squares of the differences between the measured values ${\varepsilon^{\prime }}_{d}(\omega_{k})$ and $\sigma_{d}(\omega_{k})$ and the modeled values $\varepsilon^{\prime }\left(\omega_{k}\right)$ and $\sigma \left(\omega_{k}\right)$, normalized by the measured values, summed over all measured frequencies (*k*):(9)$$Error=\sum_{k=1}^{N}(\frac{\varepsilon^{\prime }\left(\omega_{k}\right)-{\varepsilon^{\prime }}_{d}(\omega_{k})}{average\;of\;\varepsilon^{\prime }(\omega )}{)}^{2}+\sum_{k=1}^{N}(\frac{\sigma \left(\omega_{k}\right)-\sigma_{d}(\omega_{k})}{average\;of\;\sigma (\omega )}{)}^{2}$$
where
*N* is the total number of data points.The average of average of *ε*′(*ω*) and average of average of *σ*(*ω*) are the average values of the real permittivity and the conductivity over all measured frequencies.

The results of the error calculation are detailed using Equation (9), and the comparative data are displayed in Table 3. This table contrasts the performance of the four dielectric models for the different types of tumors. It is observed that the Double Cole–Cole model exhibits the least error for the four types of tumors analyzed.

So far, the results obtained provide a clear reference to the heterogeneous behavior of biological tissues, offering valuable reference data [17,18,19,20,21,22]. However, it is essential to conduct more detailed analyses for each type of tumor, as has been initiated in this study. Despite these advances, there are still various types of tumors that have not been included in the research. The inclusion of these tumors is crucial to enrich the study, enhancing its depth and accuracy. This could represent a significant advancement in clinical diagnosis, providing important contributions to medicine.

## 5. Discussion

This study represents a significant advancement in the identification and characterization of excised breast tumors using microwave dielectric spectroscopy. The obtained results show clear differences between the four types of previously excised tumors, either malignant—malignant (ductal, **ε**_r_ = 45.6; mucinous, **ε**_r_ = 59.5; lobular, **ε**_r_ = 35.1)- or benign (**ε**_r_ = 27.6) tumors. This achievement highlights the importance of this technique in fast and accurate identification of diagnosis, overcoming previous limitations of dielectric spectroscopy thanks to improved control over measurement variables.

The clinical relevance of these findings is considerable, as the identification of clear differences in permittivity between benign and malignant tumors and their typologies could revolutionize diagnostic and therapeutic approaches to breast cancer. These results enable faster and more effective clinical decision making after surgery and underscore the potential of this technique to improve the early detection of breast tumors, especially as variations in permittivity remain consistent within a specific frequency range.

Despite these advancements, the study faces challenges related to its dependence on controlled measurement variables and specific considerations for patients undergoing chemotherapy. Another point to consider is heterogeneous perfusion in the tumor. These aspects underline the importance of continuing research and refinement of techniques. This phenomenon suggests an anomaly in the formation and adaptation of blood vessels within tumors, which could be key to better understanding and treating these cancerous growths [32,33].

Thus, it can be summarized that the proposed microwave spectroscopy technique being used on excised breast tumors provides a quick and reliable method of immediately identifying the type of excised tumor. Then, the proposed microwave spectroscopy technique anticipates and complements the results latterly obtained by histological analysis and opens up a plethora of possibilities for quickly identifying the type of tumor. Nonetheless, further tests will have to be undertaken to more precisely validate the protocol and map with histological analysis.

## 6. Conclusions

In recent years, microwave dielectric characterization has emerged as a tool to complement traditional histological classification. Till now, diagnosis was exclusively achieved via biomedical methods. This paper showcases a major advancement in the microwave dielectric characterization of breast tumors using dielectric spectroscopy, clearly differentiating between benign and malignant tumors. The results show clear differences between the four types of analyzed tumors, either malignant–malignant (ductal, *ε_r_* = 45.6; mucinous, *ε_r_* = 59.5; lobular, *ε_r_* = 35.1) or benign (*ε_r_* = 27.6) tumors. Furthermore, clinically identifying permittivity differences opens up new avenues for breast cancer diagnosis and treatment without having to wait for histological analysis.

For completeness, a statistical analysis based on a double Cole–Cole model has been provided for conducting more in-depth studies, enabling the execution of specific tests for system validation. Not only does this model enhance the accuracy of simulations but also aids in gaining a more detailed understanding of the microwave dielectric characteristics of tumors. These advancements can significantly contribute to oncology, offering complementary valuable tools for effective breast cancer diagnosis and treatment.

## Figures and Tables

**Figure 1 cancers-16-00793-f001:**
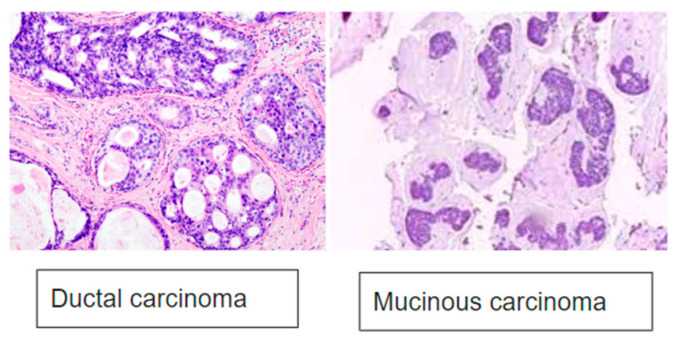
Histological types of breast tumors (taken from IRENSUR).

**Figure 2 cancers-16-00793-f002:**
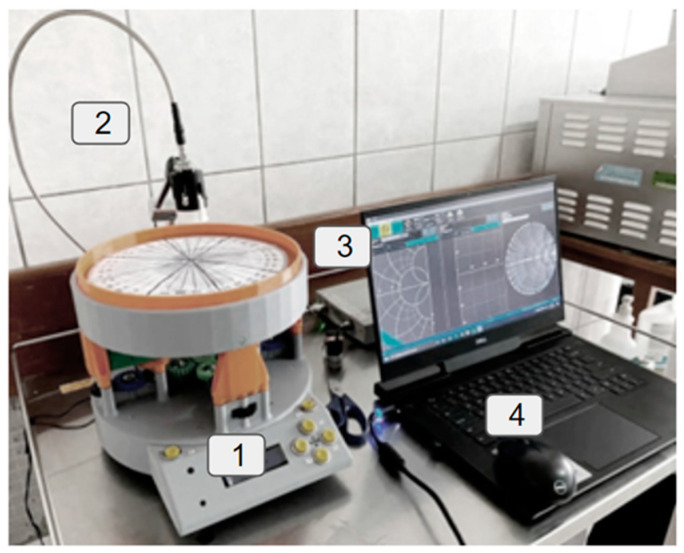
Measuring system.

**Figure 3 cancers-16-00793-f003:**
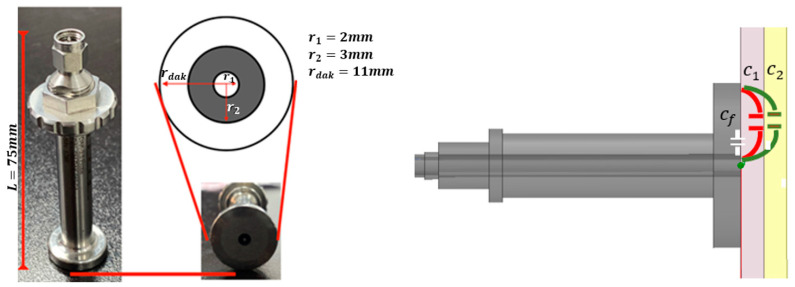
Dimensions of the used probe DAK 3.5 (TL2).

**Figure 4 cancers-16-00793-f004:**
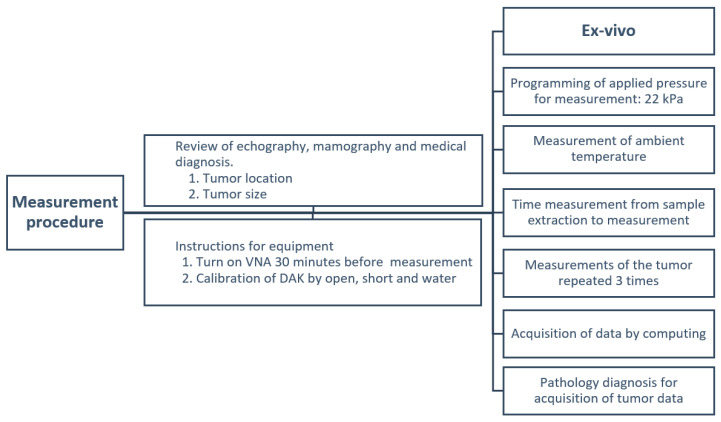
Measurement protocol.

**Figure 5 cancers-16-00793-f005:**
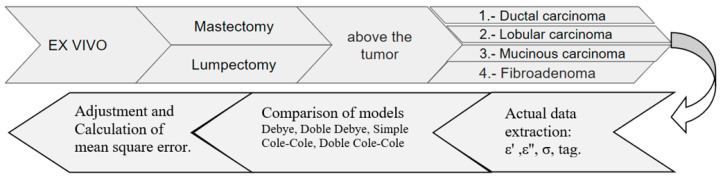
Experimental design.

**Figure 6 cancers-16-00793-f006:**
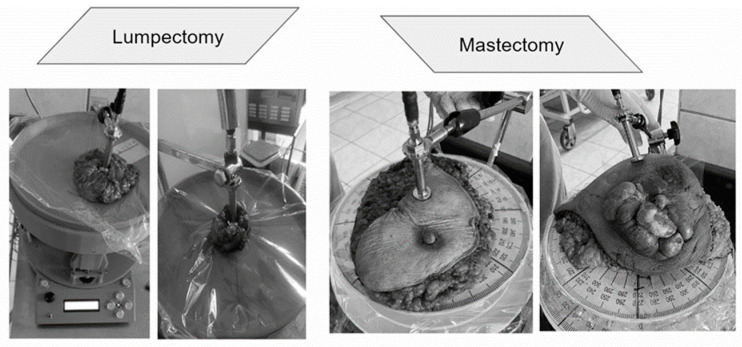
Types of surgery (taken from IRENSUR).

**Figure 7 cancers-16-00793-f007:**
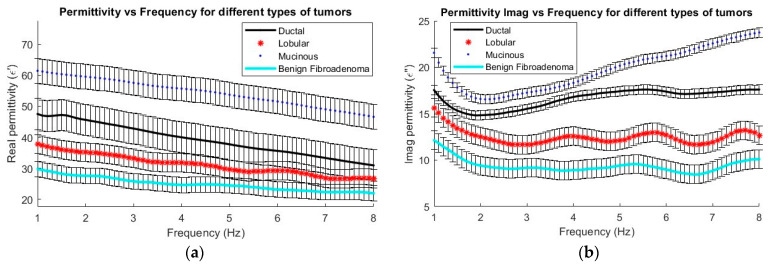
Permittivity by tumor histology: (**a**) real part and (**b**) imaginary part.

**Figure 8 cancers-16-00793-f008:**
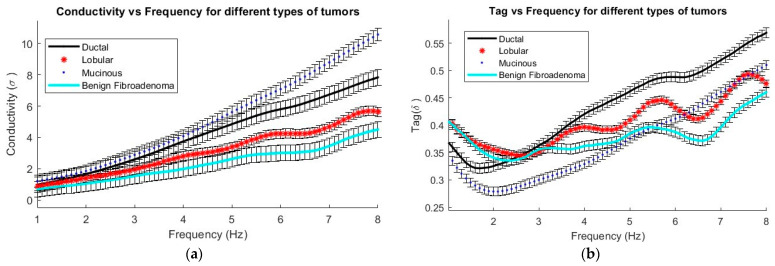
Dielectric characterization parameters extracted from permittivity: (**a**) conductivity and (**b**) loss tangent.

**Figure 9 cancers-16-00793-f009:**
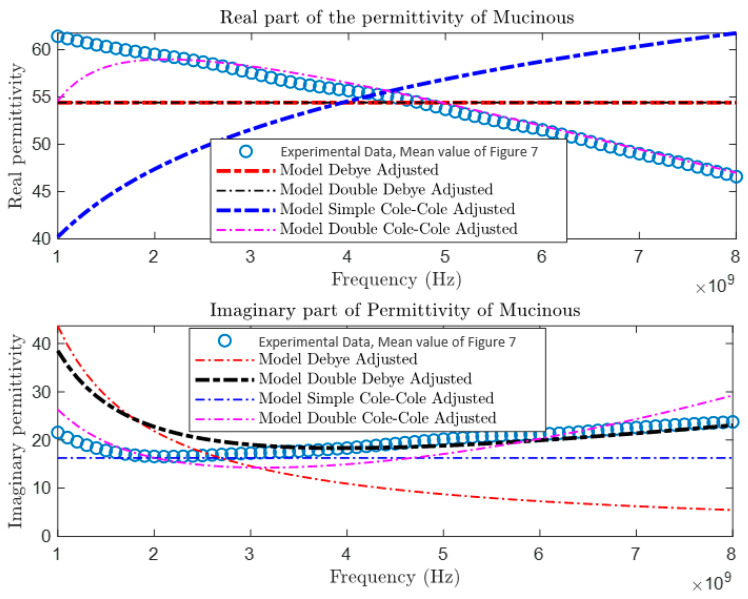
Permittivity of mucinous carcinoma.

**Figure 10 cancers-16-00793-f010:**
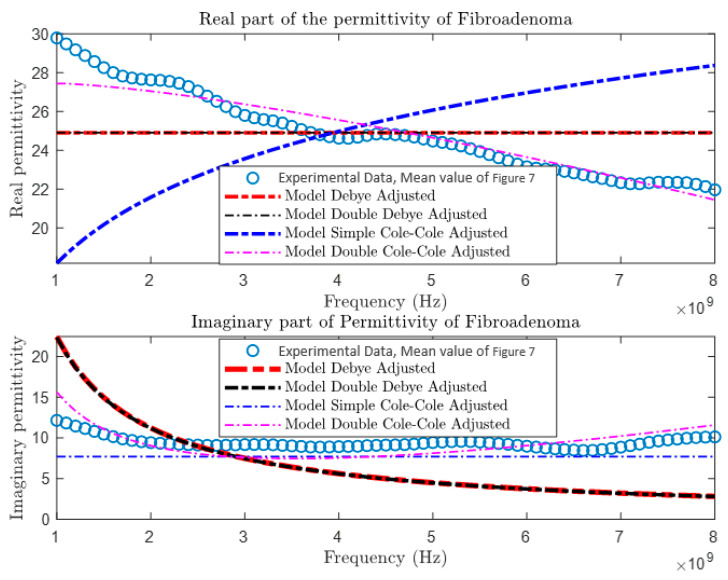
Permittivity of fibroadenoma.

**Figure 11 cancers-16-00793-f011:**
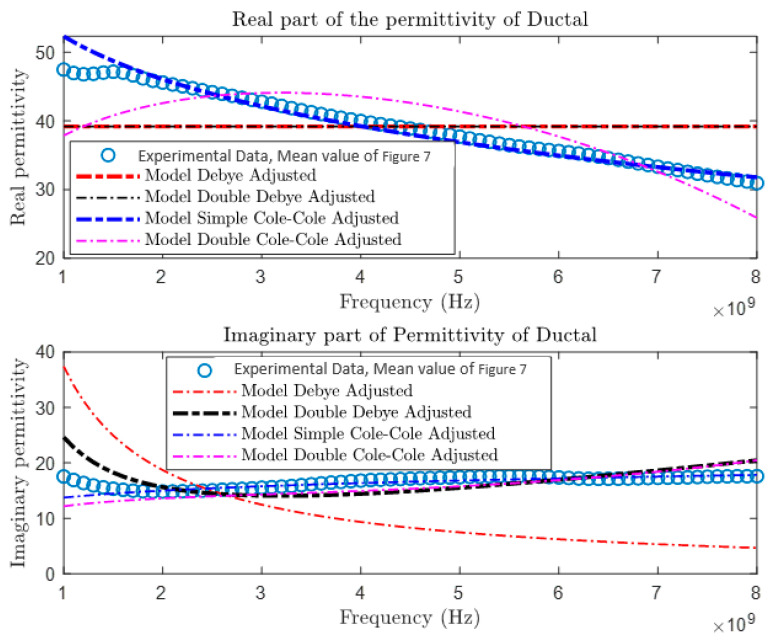
Permittivity of ductal carcinoma.

**Figure 12 cancers-16-00793-f012:**
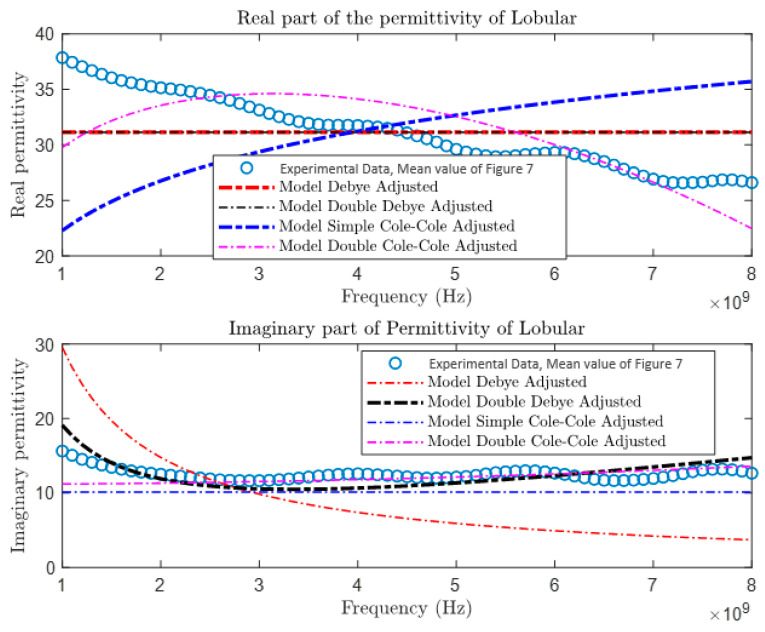
Permittivity of lobular carcinoma.

**Table 1 cancers-16-00793-t001:** Experimental data characteristics.

Tumor Type	Histological Type	N Patients	N Samples	N Measurements
Malignant	Ductal carcinoma	30	40	120
	Lobular carcinoma	10	13	39
	Mucinous carcinoma	10	13	39
Benign	Fibroadenoma	20	27	81

**Table 2 cancers-16-00793-t002:** Range of changes in permittivity level.

Tumor Type	Histological Type	N Measurements	Permittivity	
			2 GHz	4 GHz
Malignant	Ductal carcinoma	120	45.6 ± 6	39.9 ± 6
	Lobular carcinoma	39	35.1 ± 4	31.7 ± 4
	Mucinous carcinoma	39	59.4 ± 5	55.6 ± 5
Benign	Fibroadenoma	81	27.6 ± 4	24.6 ± 4

**Table 3 cancers-16-00793-t003:** Error.

Model	Mucinous	Lobular	Ductal	Fibroadenoma
Debye	26.589	13.395	13.542	27.392
Double Debye	0.074	0.432	0.192	27.393
Simple Cole–Cole	7.684	2.285	2.060	26.618
Double Cole–Cole	0.042	0.410	2.145	2.083

## Data Availability

The data presented in this study are available on request from the corresponding author.

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
