# Peer review of "Dielectric Characterization of Ex-Vivo Breast Tissues: Differentiation of Tumor Types through Permittivity Measurements"

_cancers, 2024, doi:10.3390/cancers16040793_

Round 1

Reviewer 1 Report

Comments and Suggestions for Authors

1. Introduction

The state-of-the art referenced in the Introduction of the paper is too limited and certainly not representative of the actual state-of-the art in microwave breast imaging. More specifically:

-Line 79: In addition to Refs [10,11], at least one or two review papers  should be cited. 

- Line 86: Refs [16] and [17] are supposed to cite work on breast cancer diagnosis based on dielectric properties, however Ref. [16] concerns a brain stroke detection application. Please revise.

- Line 91: To my understanding, Refs. [20-23] are not at all proper citations and do not really address the topic of bound water effects that could play a role in specific frequency ranges. Please review and revise. 

2. Materials and Methods

- Line 113: It is stated that all samples were excised, after the patients had chemotherapy treatment. In my opinion, this questions the validity and usefulness of the study results overall, given that prior chemotherapy is highly probable to have affected the consistency of the tumor tissues, and subsequently their dielectric properties. The reported results seem very interesting in terms of the potential for discrimination among the various tumour types (and especially malignant Vs benign) based on their complex permittivity values. However, would the findings really be comparable if chemotherapy had not taken place before? Are any additional support data available that could provide some information on the above? If not, I strongly believe that the study should be repeated on samples of patients who did not undergo any chemotherapy.

- Figure 4: is not easily readable. It should be re-inserted in better quality.

3. Experimental Design

- Lines 233-236: The decision on including in the study only tissue samples post-chemotherapy is justified, while recognising the notable variations attributable to chemotherapy. Is it normal that the benign tissue samples (firbroadenomas) had undergone chemotherapy treatment before? Could you please justify?

- Table 1: Please define in what patient cases more than 1 sample was taken. In case of larger tumours maybe? Please clarify the methodology and justify its consistency.

4. Results and Discussion

- Lines 291-292: "Figure 7 illustrates....": the sentence is grammatically incomplete. Please review and revise accordingly. 

- Line 317: 'probability'? It should read 'permittivity' I guess? Please revise.

- Figures 10-13: In all four figures, in the upper subplots reporting the real part of the permittivity the red curves corresponding to 'Model Debye Adjusted' do not seem to appear clearly in the graphs? Please review and revise.

- Table 3: please correct the two typos 'double' instead of 'doble'.

- Legends of Figures 10-13: Idem, please correct all the typos 'double' instead of 'doble'.

5. Conclusions

- Please correct the numbering and typo on the title of the section, which currently reads '6.  .Conclusions'.

Comments on the Quality of English Language

The paper is overall well written. Editing and some minor corrections, like the typo 'doble', instead of 'double' which appears repeatdly, should be spotted and performed.

Author Response

The answer to reviewer 1 are attached in a file.

Reviewer 2 Report

Comments and Suggestions for Authors

Thankyou for doing this intersted work. I have some questions as the follwoing: 

1-          The authors say that ‘for the first time, noticeable differences between the four types of tumors, which can 19 help to make an instant diagnosis of the excised tumor and speed up subsequent treatment. ‘ In my opinion, that is not right based in many studies since 2007.

2-          What are the advantages of using the proposed technique for measuring the dielectric peripeties of the four types of cancer and they did not measured and compared the dielectric properties of the normal tissues.   

3-          What are the advantages of using the proposed technique for measuring the dielectric peripeties in that specific range of frequencies? It seems suitable for microwave imaging approaches  and the size of the tumors is very important factor for measuring the dielectric properties?

4-          Across the frequency range of 1-8 GH. What is the difference in the dielectric properties between the tumor and normal tissues in the used frequency band?  

5-          The authors should be known there is difference between cancer /begin/ tumors in dielectric properties. They did not specify what is the type of tumors that will use their proposed technique to determine the size.

6-          What are the relationship between the tumor size and Identification sizes of the four types and their dielectrics properties and the probe  obtained from the measurement ?

7-          The authors say that” The Vector Network Analyzer (VNA) is powered on 30 209 minutes before the measurement. This ensures the equipment is stabilized and ready 210 for use” that is not right in terms of using VNA due to the affect of thermal during of using the VNA for long time.  

Comments on the Quality of English Language

It semms good 

Author Response

The answers to reviewer 2 are attached in a file.

Reviewer 3 Report

Comments and Suggestions for Authors

Peer Review Report

Manuscript ID: Cancers-2828586

Title:Dielectric Characterization of Ex Vivo Breast Tissues: Differentiation of Tumor Types Through Permittivity Measurements”

The study “Dielectric Characterization of Ex Vivo Breast Tissues: Differentiation of Tumor Types Through Permittivity Measurements” by Fernandez-Aranzamendi et al. focuses on evaluating the dielectric characteristics of excised human breast tissues particularly four breast tumors (ductal carcinoma, lobular carcinoma, mucinous carcinoma, fibroadenoma). The study cannot be accepted in present form and therefore, we propose major revision while giving authors an opportunity to further improve the work while discussing their results. The authors must thrust on logical and quantitative discussion with qualitative inferences, while also considering comparing and contrast with literature. The global perspective was missing, and all dots of the work must connect to conclude meaningful inferences. The authors must state that if they have used any artificial intelligence assisted tools to write, extract or infer any particular section of this work.

1. The authors should re-write Simple Summary in a language used in Professional Practice. For example: (Lines 14-18)

For the first time, and thanks to the help of 70 volunteered patients who already had chemotherapy treatment, an accurate tissue characterization of the breast tumors, benign or malignant (Ductal carcinoma, Lobular carcinoma, Mucinous carcinoma, Fibroadenoma) in terms of relative electric permittivity and

loss tangent has been undertaken.

2. The writing is unclear to me and should be re-written. (Lines 18-20)

The obtained results on both, relative permittivity and loss tangent, show, for the first time, noticeable differences between the four types of tumors, which can help to make an instant diagnosis of the excised tumor and speed up subsequent treatment.

3. The Title Should not be capitalized (not Upper-Case).

4. The authors should make the usage of word EX VIVO as italic with an hyphen (ex-vivo).

5. The authors should rewrite the Abstract. There are technical jargons along with unstructured and very long sentences which must be shortened and revised.

The main objective is to identify significant differences between each type of ex vivo breast tissue and define decision ranges among them. Utilizing an open-ended coaxial probe in a 1-8 GHz frequency range, an experiment that classified data from 70 patients into four tissue types has allowed us to make important conclusions on early diagnosis.

6. Line 30 and 31: Please note you have put commas instead of period (.). For example: 33.1 for lobular, 59.5 for mucinous, and 27.6 for benign tumors. Table 2 has similar problems. Line 407-408.

7. Ensure that while you embed references, it must appear to be together rather than separate. For example: Line 91: [20-23]. Implement such changes throughout the manuscript.

8. In the Introduction, the authors must classify different types of breasts before moving to explain four different tumors [Estimation of Clinical size of breast tumour lesions using contrast enhanced magnetic resonance imaging: Delineation of tumour boundaries]. Link this comment with Lines 146-149 in your manuscript.

9. The authors haven’t discussed the blood perfusion heterogeneity that gives rise to different dielectric properties. The authors should discuss the heterogeneity of blood perfusion [Modified Pennes bioheat equation with heterogeneous blood perfusion: A newer perspective] in Introduction as well as Discussion section.

10. What is the impact of temperature on dielectric properties of the sample? How would it affect your measurements if the properties were measured in relatively cool or relatively warm conditions? I suspect that permittivity/conductivity might be affected by external environment stressors.

11. What does the bars represent in Fig. 7?

12. How would the permittivity and conductivity change with temperature? Plot.

13. Why there are no statistical deviations in the experimental data? Please provide such error bars? (Figs. 10-13)

14. There must not be dot (.) before Conclusions in Line 402?

15. Line 392 should be expanded in reference to [Modified Pennes bioheat equation with heterogeneous blood perfusion: A newer perspective].

16. The authors must report their results within two standard errors of mean.

17. Rewrite Conclusion section with meaningful inferences.

18. Discussion should be elaborated by drawing meaningful inferences while comparing and contrasting the true efficacy of the results.

General Comments:

Comment #1: One of the primary concerns with this paper is the lack of clarity in defining the scope and objectives of the study. The authors have not clearly outlined the specific research questions or goals they aimed to address. As a result, the paper appears to be a collection of loosely connected ideas rather than a cohesive discussion.

Comment #2: Another major issue is the lack of rigorous methodology. The paper does not provide any information regarding the systematic studies used to perform this work.

Comment #3: The paper also suffers from a lack of critical analysis and evaluation of the reviewed technologies. Merely describing and summarizing the existing technologies without offering any meaningful insights or comparisons diminishes the value of the work. The paper should have critically examined the strengths, limitations, and potential areas of improvement for each approach.

Comment #4: The overall organization and structure of the paper are inadequate. The flow of ideas is unclear, and there is a lack of coherence between sections. The paper should have presented a clear introduction, outlined the main themes or categories of technologies, and provided a concise summary or conclusion to tie the information together.

Comment #5: Spacing issues. Correct such issues throughout the manuscript.

Comment #6: Abbreviations must be explained next to the first mention or wherever appropriate to avoid reader struggle understanding the terminology. We have identified technical jargons which must be avoided by the authors.

Comment #7: Unrequired abbreviations otherwise should be discarded.

We are looking forward to reviewing your next manuscript with addition of all necessary details recommended by the reviewer.

Comments on the Quality of English Language

The authors must revisit the manuscript as there are very long and unstructured sentences throughout the manuscript.

Author Response

The answers to reviewer 3 are attached in a file.

Round 2

Reviewer 2 Report

Comments and Suggestions for Authors

Great revsion 

Comments on the Quality of English Language

Great revsion 

Reviewer 3 Report

Comments and Suggestions for Authors

Please re-review for typographical errors. Why there is no deviation in two standard errors of mean?

Comments on the Quality of English Language

Avoid technical jargons in your writing.